# Social determinants of the changing tuberculosis prevalence in Việt Nam: Analysis of population-level cross-sectional studies

Nicola Foster[1,2]*, Hai V. Nguyen[3,4,5], Nhung V. Nguyen[3], Hoa B. Nguyen[3],
Edine W. Tiemersma[6], Frank G. J. Cobelens[4,5], Matthew Quaife[1,2],
Rein M. G. J. Houben[1,2]

**1** TB Modelling Group, Department of Infectious Disease Epidemiology, London School of Hygiene &
Tropical Medicine, London, United Kingdom, **2** TB Centre, London School of Hygiene & Tropical Medicine,
London, United Kingdom, **3** Việt Nam National Tuberculosis Programme, Hanoi, Việt Nam, **4** Department of
Global Health, Amsterdam University Medical Centers, Amsterdam, the Netherlands, **5** Amsterdam Institute
for Global Health and Development, Amsterdam, the Netherlands, **6** KNCV Tuberculosis Foundation, the
Haag, the Netherlands

* nicola.foster@lshtm.ac.uk

pmed.1003935

Medicine Editorial Board, UNITED STATES

**Data Availability Statement:** Data analysed in this
study was provided by the Việt Nam National Lung
Hospital, subject to the signing of an agreement

## Abstract

### Background

An ecological relationship between economic development and reduction in tuberculosis
prevalence has been observed. Between 2007 and 2017, Việt Nam experienced rapid eco-
nomic development with equitable distribution of resources and a 37% reduction in tubercu-
losis prevalence. Analysing consecutive prevalence surveys, we examined how the
reduction in tuberculosis (and subclinical tuberculosis) prevalence was concentrated
between socioeconomic groups.

### Methods and findings

We combined data from 2 nationally representative Việt Nam tuberculosis prevalence sur-
veys with provincial-level measures of poverty. Data from 94,156 (2007) and 61,763 (2017)
individuals were included. Of people with microbiologically confirmed tuberculosis, 21.6%
(47/218) in 2007 and 29.0% (36/124) in 2017 had subclinical disease. We constructed an
asset index using principal component analysis of consumption data. An illness concentra-
tion index was estimated to measure socioeconomic position inequality in tuberculosis prev-
alence. The illness concentration index changed from −0.10 (95% CI −0.08, −0.16; $p$ =
0.003) in 2007 to 0.07 (95% CI 0.06, 0.18; $p$ = 0.158) in 2017, indicating that tuberculosis
was concentrated among the poorest households in 2007, with a shift towards more equal
distribution between rich and poor households in 2017. This finding was similar for subclini-
cal tuberculosis. We fitted multilevel models to investigate relationships between change in
tuberculosis prevalence, individual risks, household socioeconomic position, and
neighbourhood poverty. Controlling for provincial poverty level reduced the difference in
prevalence, suggesting that changes in neighbourhood poverty contribute to the explanation
of change in tuberculosis prevalence. A limitation of our study is that while tuberculosis

that the data is kept confidential and is not made available to others. Researchers wishing to use the data should apply to Dr Hoa and the Institutional Research Board at the Việt Nam National Lung hospital by emailing bvptw@bvptw.org. A description of the dataset, and an overview of the variables analysed plus the code required to produce the analysis may be found at https://doi.org/10.17037/DATA.00002373.

**Funding:** RMGJH and NF were supported by a European Research Council starting grant (TBornotTB, action number 757699) to conduct the analysis presented here. The funders had no role in study design, data collection and analysis, decision to publish, or preparation of the manuscript.

**Competing interests:** The authors have declared that no competing interests exist.

**Abbreviations:** AWE, absolute wealth estimate; GDP, gross domestic product; LJ, Löwenstein–Jensen; MLM, multilevel model; PR, prevalence ratio; SEP, socioeconomic position.

prevalence surveys are valuable for understanding socioeconomic differences in tuberculosis prevalence in countries, given that tuberculosis is a relatively rare disease in the population studied, there is limited power to explore socioeconomic drivers. However, combining repeated cross-sectional surveys with provincial deprivation estimates during a period of remarkable economic growth provides valuable insights into the dynamics of the relationship between tuberculosis and economic development in Việt Nam.

## Conclusions

We found that with equitable economic growth and a reduction in tuberculosis burden, tuberculosis became less concentrated among the poor in Việt Nam.

## Author summary

### Why was this study done?

- Historically, large reductions in tuberculosis prevalence globally have been ascribed to changes in living standards, such as housing and nutrition, that come with economic development.

- Previous studies have shown that social protection policies (a component of economic development) may reduce tuberculosis incidence, but that such gains are dependent on the amount invested in social protection policies.

- However, direct evidence of the interaction between economic growth and tuberculosis burden is limited, and evidence is missing with regards to equity.

### What did the researchers do and find?

- We used data from consecutive tuberculosis prevalence surveys conducted during a time of rapid economic growth in Việt Nam to analyse the association between equitable economic development and reduction in tuberculosis prevalence.

- We found a significant shift in the distribution of tuberculosis from disproportionately affecting poor households towards a more equitable distribution of the reduced tuberculosis prevalence among the population, closely linked to neighbourhood poverty indicators.

### What do these findings mean?

- Our work contributes to the body of evidence of social determinants of tuberculosis prevalence.

- A more equitable burden of tuberculosis disease is possible in the context of rapid, and equitable, economic growth.

- Further work is required to understand how improvements in healthcare services contribute to or mediate the drive towards a more equitable burden of tuberculosis.

## Introduction

The relationship between tuberculosis disease and economic development is well documented [1–3], while the association between subclinical tuberculosis and economic development has received comparatively little attention to date [4]. Ecological studies analysing historical trends have attributed sustained reductions in the prevalence of tuberculosis to a combination of reductions in crowded living conditions [5], effective anti-tuberculosis chemotherapy [6], and the improvements in housing, access to health services, and nutrition that accompany economic development, poverty reduction, and social policies such as social protection [7]. The World Health Organization's End TB Strategy and the UN's Sustainable Development Goals 1 and 3 recognise the importance of healthcare and the control of communicable diseases, including tuberculosis, as outcomes of, and crucial contributors to, economic development [8–10].

Recent empirical work has attempted to quantify the effect of social protection, as an intervention to reduce poverty, on programmatic indicators such as tuberculosis prevalence and case detection rates [1,7,11–16]. Carter et al. considered how social protection, as a component of economic development policy, may affect tuberculosis incidence [11]. Social protection refers to policies designed to reduce poverty through improvements in the labour market, and support for poor and sick individuals. They found that social protection may reduce the incidence of tuberculosis by 76% [11]. In evaluating the relationship between social protection and economic development, Siroka et al. found that tuberculosis prevalence is reduced with increased spending on social protection, though this effect plateaued when countries spent more than 11% of gross domestic product (GDP) on social protection [14]. Although these studies provided evidence that economic growth and social protection are associated with reductions in tuberculosis burden, they did not explore how the distribution of tuberculosis prevalence changes during economic growth.

Việt Nam is an example of a country that has experienced notable sustained economic growth. In 2006, the smear-positive tuberculosis incidence in Việt Nam was estimated to be 260 per 100,000 population, and the treatment success rate was 92% [17]. National tuberculosis prevalence surveys were conducted in Việt Nam in 2007 and 2017 [18,19]. When differences in tuberculosis screening and diagnostic practices were accounted for, a comparative study showed a decline in tuberculosis prevalence over the 10-year period [20]. The study found a 37% reduction in the prevalence of culture-positive tuberculosis, a 53% reduction in the prevalence of smear-positive tuberculosis, and no significant reduction in smear-negative or subclinical tuberculosis. The change in tuberculosis prevalence was more pronounced among men, among people living in rural areas, and in provinces in the north and south of the country [20].

In 1986 a series of economic reforms, the Di Mi Policy, were introduced that included investments in health and education [21]. Since then, Việt Nam has experienced rapid and sustained economic growth, with GDP per capita rising from US$230 in 1985 to US$906 in 2007 and US$2,343 in 2017. During this period, income inequality as measured by the Gini coefficient has remained stable for over a decade (35.8 in 2006 and 35.7 in 2018) [22,23]. The

increase in GDP per capita with an unchanging Gini coefficient suggests that the benefits of the rapid economic development observed in Việt Nam have been distributed equitably among the population, an example of shared prosperity. Given that the individual risk of tuberculosis disease is increased by poor household socioeconomic position (SEP), the rapid and sustained economic growth in Việt Nam was an opportunity to examine simultaneous changes in tuberculosis prevalence and economic growth.

In the analysis presented here, we used the opportunity of measured longitudinal changes in both poverty and tuberculosis burden to estimate the differential prevalence of tuberculosis by SEP and to examine the individual, household, and neighbourhood social determinants of the reduction in tuberculosis prevalence in Việt Nam.

## Methods

We combined individual-level data from 2 cross-sectional nationally representative tuberculosis prevalence surveys to measure the social determinants of changes in tuberculosis prevalence in Việt Nam [18,19]. The SEP of households was estimated by constructing an asset index from consumption data; illness concentration curves and an illness concentration index represented the distribution of illness. Associations between tuberculosis prevalence, individual risk factors, and household SEP within neighbourhoods were estimated by fitting mixed effects multilevel models (MLMs) [24–26].

### Possible causal pathways

Fig 1 shows the causal model for the analysis arranged by individual risks and household and neighbourhood effects [27]. Causal models are representations of assumed causal structures and provide a framework for discussing study design, variables included and how this may affect our understanding of the measure of interest [28].

Social determinants of health are the socioeconomic, cultural, and political aspects of the community that affect the health of populations [29]. These determinants include people's living and working conditions, water and sanitation, housing, unemployment, and political drivers of health. An individual's biological risk of developing tuberculosis is influenced by age, gender, and previous treatment and how these intersect with household risk and household economic position in the neighbourhood [30]. Transmission of tuberculosis is spatially concentrated in neighbourhoods [16]. Similarly, economic development leads to increased opportunities in neighbourhoods, and depending on how wealth is distributed, there may be a reduction in unemployment, greater assistance to households in need, and therefore more resources per capita. Equitable economic development improves neighbourhood economy, which improves living conditions through reduced crowding, increased availability of windows to improve air circulation, and reduced periods of transmission in neighbourhoods [31]. Furthermore, improvements in the neighbourhood economy increase household resources, reducing malnutrition and improving households' ability to seek healthcare [1]. Comparatively wealthier households would have greater ability to negotiate access to neighbourhood resources such as housing and health services, therefore lowering their risk of tuberculosis. If symptomatic (clinical tuberculosis), individuals would be more likely to seek and receive tuberculosis care, reducing transmission periods. However, if not symptomatic (subclinical), diagnosis within health services focused on passive tuberculosis case finding may be delayed until onset of clinical disease, leading to increased tuberculosis prevalence in the population [4,32].

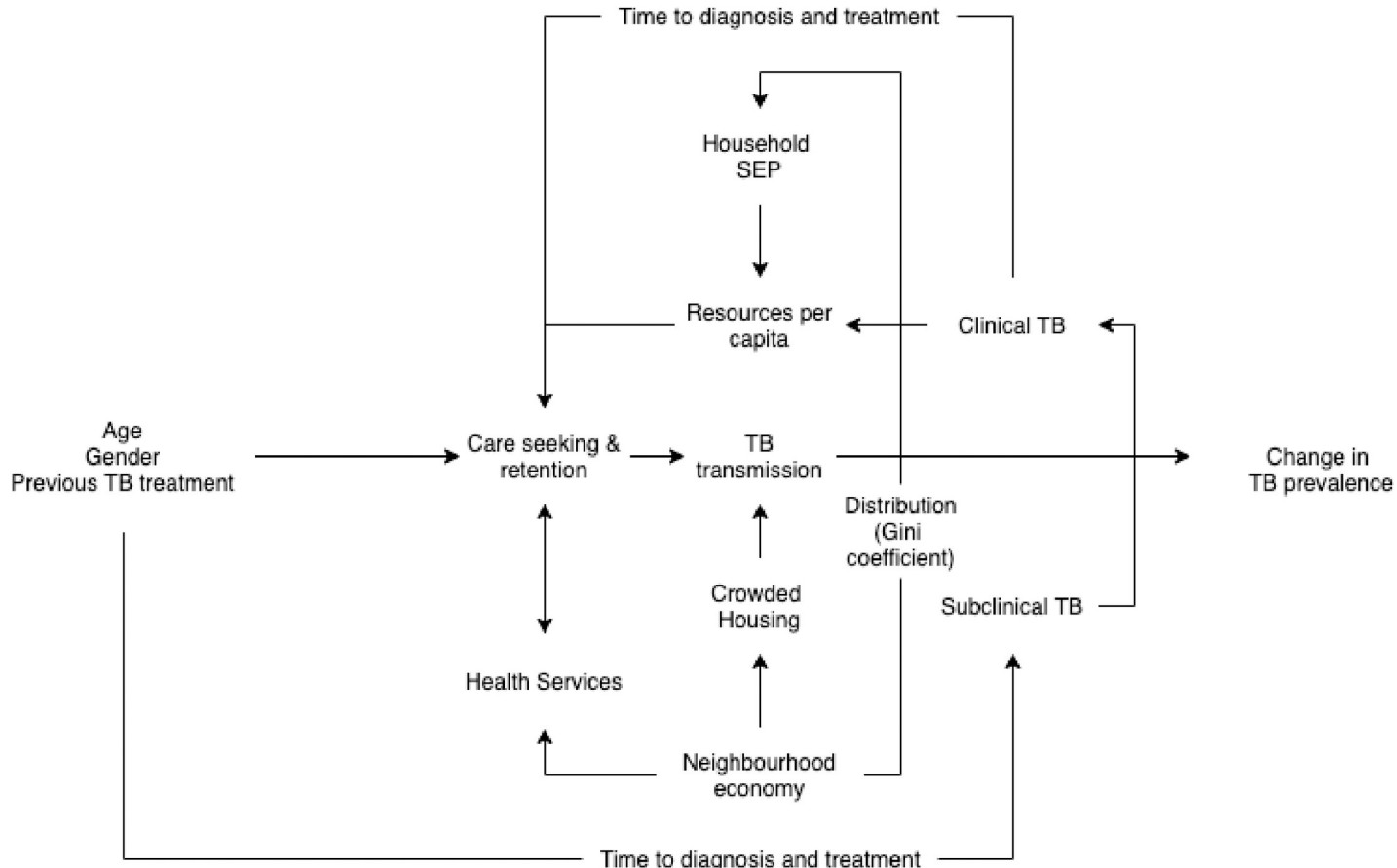

**Fig 1. Causal diagram of social determinants of tuberculosis prevalence in Việt Nam.** SEP, socioeconomic position; TB, tuberculosis.

### Việt Nam national tuberculosis prevalence surveys and case definitions

Nationally representative Việt Nam tuberculosis prevalence surveys were conducted in 60 of 63 provinces in 2007 and 2017 [18,19]. A detailed explanation of the research procedures and differences between the 2 surveys are published elsewhere [20]. In summary, individuals were identified for inclusion in the surveys by multistage sampling whereby first districts and then communes were selected proportional to population size. Clusters (geographical sub-communes) were selected by random sampling, and all households in the selected sub-communes (70 sub-communes in survey 1, and 82 in survey 2) were included. Individuals were eligible for screening if they were 15 years of age or older. Screening procedures included questions on cough and treatment history followed by chest radiography, sputum smear microscopy, and solid Löwenstein–Jensen (LJ) culture. Individuals reporting a cough for at least 2 weeks, haemoptysis, or previous tuberculosis treatment, or who had an abnormal chest X-ray, were considered screen positive. In the first survey, 8.0% (7,529/94,156) of respondents screened positive, compared to 7.4% (4,595/61,763) in the second survey [20]. A flow diagram of participant selection is included in S1 Text.

There were improvements in diagnostic technology between the 2 surveys. For comparability, an individual was considered to have microbiologically confirmed tuberculosis if they were screen positive, had a smear microscopy test, and had at least 1 positive LJ culture. Individuals were considered to have subclinical tuberculosis if they had not reported any symptoms but had at least 1 positive LJ culture.

Data from the prevalence surveys were matched to provincial-level measures of poverty using data from the World Bank, the percentage of people living on less than US$2 per day in 2009, and the 2013 Ministry of Labour, Invalids and Social Affairs (MOLISA) metric [22,23]. The MOLISA metric is used for determining eligibility for the national anti-poverty programme and uses income as an indicator.

## Statistical analysis

The asset index was calculated using principal component analysis of 6 variables: the presence of clay floors in the home, wood used as fuel for cooking, and ownership of a stereo system, television, motorbike, and car. In the 2017 survey, the presence of a fridge, computer, air conditioner, washing machine, and water heater were also included in the survey. We restricted the asset index to the same 6 consumption variables in 2007 and 2017 [33]. Using the index, households were divided into groups of relative wealth (SEP groups), and disease prevalence was compared between these groups. We assigned consumption data responses as provided by the self-declared head of the household to all members of the household. To adjust for the relative sampling probability of each participant, we used survey sampling weights based on age, gender, cluster size, geographical area, and post-stratification adjustment. Data were analysed using Stata 16.1 and RStudio 1.3.1093 [34].

The distribution of disease between SEP groups is represented by constructing illness concentration curves [35]. These are used to quantify whether inequality in SEP exists for a health sector variable, such as tuberculosis prevalence [36]. We then quantified the position of the geometric mean on the curve by estimating the concentration index, which is defined as twice the area between the concentration curve and the line of equality (the 45-degree line on the graph) [37].

The relationships between tuberculosis prevalence and subclinical tuberculosis prevalence and SEP are explained not only by individual-level risks, but also by interactions between hierarchical levels including the household and the wider neighbourhood. In our analyses, we investigated the association between the change in tuberculosis prevalence, relative household SEP, and absolute provincial poverty [38]. We used log-binomial models to examine dependencies between variables nested in each group. We used MLMs with group- and individual-level intercepts as random effects. MLMs aim to explain the change in tuberculosis prevalence over time while considering that poverty and the risk of contracting tuberculosis are clustered geographically and in households. MLMs allow us to analyse how neighbourhood effects explain variation in change in tuberculosis prevalence over time.

By partially pooling varying coefficients, we quantified the relationship between variables where we expected the coefficients to vary between neighbourhoods. The Hausman test was used to test the correlation between random error and individual effects (regressors) in the model (see S1 Text).

This study is reported as per the Strengthening the Reporting of Observational Studies in Epidemiology (STROBE) guideline (S1 STROBE Checklist).

## Ethics

The Việt Nam national tuberculosis prevalence surveys were approved by the National Hospital of Tuberculosis and Respiratory Diseases in Hanoi (2007) and the institutional review board of the Việt Nam National Lung Hospital (2017; 62/17/CTHKH). This analysis was approved by the ethics committee of the London School of Hygiene & Tropical Medicine (16396).

## Results

The characteristics of study participants are summarised in Table 1. Data from 155,919 participants were included in the study, 94,156 from survey 1 and 61,763 from survey 2, of which 0.23% (218/94,156) in survey 1 and 0.20% (124/61,763) in survey 2 had microbiologically confirmed tuberculosis. Of the patients with microbiologically confirmed tuberculosis, 21.6% (47/218) in survey 1 and 29.0% (36/124) in survey 2 reported no cough and were therefore considered to have subclinical tuberculosis. The average age of study participants was 40.1 and 46.6 years old, respectively. The gender balance was similar between the 2 surveys, with 54.8% (51,560/94,156) of survey 1 participants and 56.0 (34,613/61,763) of survey 2 participants being male. Similar proportions of patients in the 2 surveys reported at least 1 tuberculosis-associated symptom: 21.7% (20,474/94,156) in survey 1 and 19.3% (11,917/61,763) in survey 2,

**Table 1. Comparison of the characteristics of study participants between survey 1 and survey 2.**

| Characteristic | Survey 1 (2007) | | Survey 2 (2017) | |
|---|---|---|---|---|
| | Percent | *n/N* participants or mean (SD) | Percent | *n/N* participants or mean (SD) |
| Microbiologically confirmed tuberculosis | 0.23% | 218/94,156 | 0.20% | 124/61,763 |
| Microbiologically confirmed tuberculosis—subclinical | 21.6% | 47/218 | 29.0% | 36/124 |
| **Individual** | | | | |
| Age (years) | | | | |
| 15–24 | 22.2% | 20,934/94,156 | 10.6% | 6,542/61,763 |
| 25–34 | 19.8% | 18,681/94,156 | 16.5% | 10,191/61,763 |
| 35–44 | 21.0% | 19,790/94,156 | 18.6% | 11,508/61,763 |
| 45–54 | 17.3% | 16,285/94,156 | 21.5% | 13,289/61,763 |
| 55–64 | 8.6% | 8,138/94,156 | 18.0% | 11,143/61,763 |
| ≥65 | 11.0% | 10,328/94,156 | 14.7% | 9,090/61,763 |
| Gender | | | | |
| Male | 54.8% | 51,560/94,156 | 56.0% | 34,613/61,763 |
| Female | 45.2% | 42,596/94,156 | 44.0% | 27,150/61,763 |
| Of all participants, those with at least 1 tuberculosis-associated symptom | 21.7% | 20,474/94,156 | 19.3% | 11,917/61,763 |
| Previous tuberculosis treatment | 1.3% | 1,228/94,156 | 1.3% | 789/61,763 |
| **Household** | | | | |
| Absolute wealth estimate | | US$2,403.80 (US$27) | | US$2,399.60 (US$26) |
| Household socioeconomic position | | | | |
| Lowest | 24.9% | 22,677/90,975 | 35.1% | 19,739/56,260 |
| Lower middle | 34.5% | 31,419/90,975 | 25.3% | 14,207/56,260 |
| Upper middle | 16.8% | 15,284/90,975 | 22.7% | 12,777/56,260 |
| Highest | 23.7% | 21,595/90,975 | 17.0% | 9,537/56,260 |
| Region | | | | |
| North | 48.5% | 45,669/94,156 | 41.4% | 25,575/61,763 |
| Centre | 15.6% | 14,646/94,156 | 21.9% | 13,525/61,763 |
| South | 35.9% | 33,841/94,156 | 36.7% | 22,663/61,763 |
| Type of residence | | | | |
| Urban | 28.0% | 26,353/94,156 | 30.2% | 18,656/61,763 |
| Remote | 29.2% | 27,532/94,156 | 25.7% | 15,882/61,763 |
| Rural | 42.8% | 40,271/94,156 | 44.1% | 27,225/61,763 |
| **Province** | | | | |
| Provincial poverty headcount percent (2009) | | 22.0 (14.6) | | 21.6 (15.9) |

*n*, sample size; *N*, population size; SD, standard deviation.

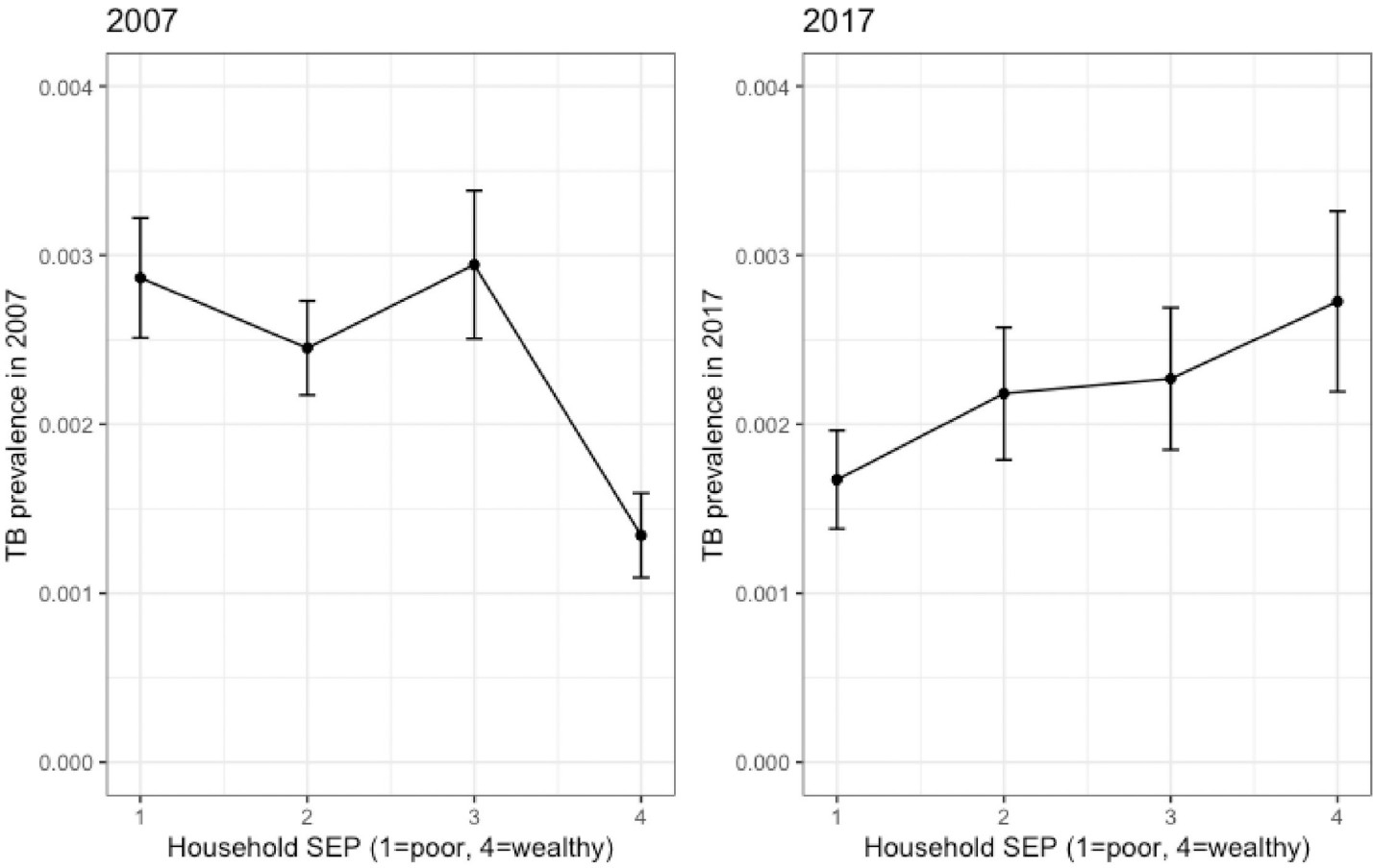

**Fig 2. The distribution of tuberculosis (TB) prevalence by socioeconomic position (SEP) as measured in the 2007 and 2017 tuberculosis prevalence surveys.** The plot shows the average household asset index and the confidence intervals around the mean.

and the proportions with previous tuberculosis treatment were similar, 1.3% (1,228/94,156) in survey 1 and 1.3% (789/61,763) in survey 2.

When comparing household SEP between surveys 1 and 2, a greater proportion of households were in the lowest SEP category (35.1%; 19,739/56,260) in the 2017 compared to the 2007 survey (24.9%; 22,677/90,975). This measure is not consistent with the absolute wealth estimate (AWE), which is similar between the 2 surveys. The AWE per household is based on the household SEP, country measures of production, and the distribution of wealth between rich and poor individuals. Therefore, these measures are related, but the AWE can be compared between time periods. The mean AWE for survey 1 was US$2,403.80 (SD US$27.00) and for survey 2 was US$2,399.60 (SD US$26.00).

The proportion of households sampled from the central region of Việt Nam was slightly larger in survey 2 compared to survey 1 (21.9% versus 15.6%), and survey 2 included more urban (30.2% versus 28.0%) and rural areas (44.1% versus 42.8%) than remote areas (25.7% versus 29.2%). The percentage of households below the poverty line (living on less than US$2 per day) was 22.0% (SD 14.6%) in 2007 compared to 21.6% (SD 15.9%) in 2017.

Fig 2 shows the proportion of study participants with microbiologically confirmed tuberculosis by SEP for each of the surveys (2007 and 2017). A shift in the distribution of tuberculosis disease from a left-leaning slope, where the disease is concentrated among poor households, to a right-leaning slope (concentrated among the wealthy) is observed. The proportion of

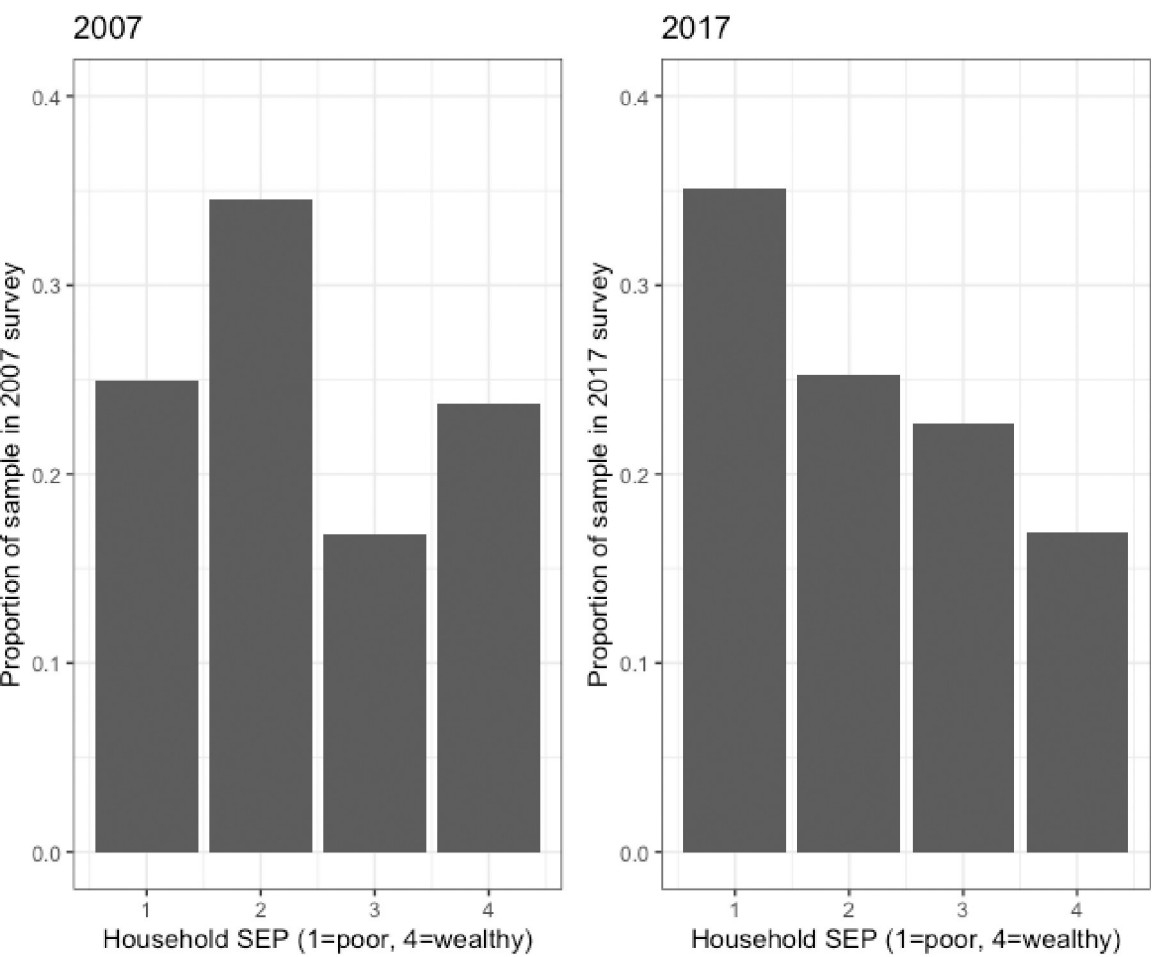

**Fig 3. Proportion of participants by socioeconomic position (SEP) in 2007 and 2017.**

participants from each of the surveys who are represented by each of the SEP groups is shown in Fig 3. In 2007, there was a similar proportion of households in the lowest and highest SEP groups. In contrast, in 2017, a greater proportion of households were categorised based on their consumption data as poor rather than wealthy.

In Fig 4, we show illness concentration curves, which represent the cumulative tuberculosis prevalence ordered by SEP, relative to the equal distribution line. In the 2007 survey, the concentration curve for tuberculosis prevalence lies above the equal distribution line; therefore, tuberculosis prevalence was concentrated among poorer households in the 2007 survey. In the 2017 survey, the concentration curve lies below the equal distribution line, indicating that tuberculosis prevalence was more equitably distributed among the population, with a higher concentration of tuberculosis in wealthier patients. These results are supported by the estimates of the illness concentration index (see Fig 5). The illness concentration index for tuberculosis disease was −0.10 (95% CI −0.08, −0.16; $p = 0.003$) in 2007 and 0.066 (95% CI 0.06, 0.18; $p = 0.158$) in 2017. When we restrict the case definition to subclinical tuberculosis, we see similar results, though with a more pronounced shift towards the wealthier households in 2017.

In Table 2, the results of evaluations of the associations between tuberculosis prevalence and individual and household risks for each survey are shown separately. In the 2007 survey, we

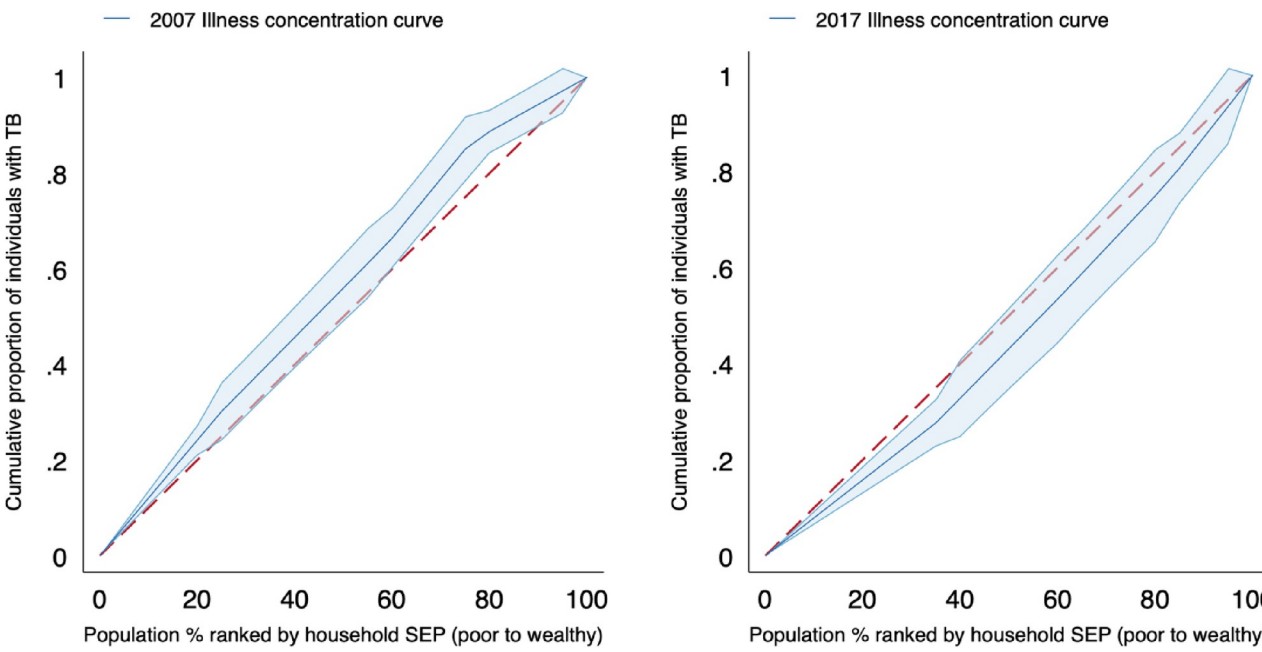

**Fig 4. Illness concentration curves.** The red dashed line represents the equal distribution line, while the blue curve is the cumulative tuberculosis (TB) prevalence in the population ranked by household socioeconomic position (SEP). The blue shaded area is the uncertainty interval. A curve above the equal distribution line means that TB is concentrated among poor households, and a curve below the equal distribution line means that TB is concentrated among wealthy households. Concentration curves for TB-associated symptoms are included in S1 Text.

found that older age (prevalence ratio [PR] = 2.79; 95% CI 2.09, 3.49; $p < 0.001$) and male gender (PR = 1.61; 95% CI 1.29, 1.92; $p < 0.001$) were associated with increased tuberculosis prevalence. Living in a remote area was negatively associated with tuberculosis prevalence in 2007 (PR = −0.16; 95% CI −0.53, 0.20; $p = 0.387$) and in 2017 (PR = −0.09; 95% CI −0.46, 0.29; $p = 0.644$). The wealthiest households were less likely (PR = −0.41; 95% CI −0.81, −0.00; $p = 0.048$) to have tuberculosis than the poorest households in 2007. These associations were similar in direction in the 2017 survey, except for the association with household SEP, where the wealthiest participants were more likely to have tuberculosis (PR = 0.76; 95% CI 0.76, 1.16; $p < 0.001$).

The results of the MLMs are shown in Table 3. We present the results of 3 mixed effects models with random intercepts. Model A is used to investigate the association between individual characteristics and tuberculosis prevalence while controlling for time and the provincial poverty rate, which is the measured as the percentage of the population below the poverty line. In Model B, we control for the district poverty rate as well as the absolute wealth of the household. In Model C, the district and relative SEP of households are controlled for to understand how individual and household risks explain the change in tuberculosis prevalence between the 2 surveys. We find that the difference in tuberculosis prevalence over time (effect size) reduces when we include indicators of household SEP and provincial poverty, which suggests that some of the observed change in tuberculosis prevalence can be explained by changes in provincial poverty.

## Discussion

We found that in the context of rapid economic growth and equitable distribution of resources in Việt Nam, there was a shift in the distribution of tuberculosis from being concentrated among poor households to a more equal distribution among households of different SEP. In the 2007 survey, older age, being male, and living in an urban centre was associated with increased tuberculosis prevalence. Conversely, in the 2017 survey, the association between

older age and tuberculosis prevalence decreased with urban living. MLMs showed the importance of provincial poverty in explaining some of the change in tuberculosis prevalence observed. Similar results were found when restricting the analysis to subclinical tuberculosis.

Studies investigating the association between reductions in tuberculosis incidence and economic development have been conducted in a range of settings [1,6,7,11,12,14]. Relationships between economic development and tuberculosis prevalence are challenging to examine given distal relationships that influence the causal pathway. Economic development may be measured as an increase in country GDP, which represents market productivity, but this is only one aspect of economic development. If economic development increases wealth inequality in a population, patients' vulnerability to tuberculosis disease may increase [1]. The role of improved healthcare in mediating that relationship is unclear. In a multi-country analysis, Dye et al. showed that rates of decline in tuberculosis incidence were associated with biological, social, and economic determinants [7]. Focusing on poverty alleviation and social protection policies, Carter et al. found that reducing extreme poverty may reduce the global incidence of tuberculosis by 33%; simultaneously expanding social protection coverage may reduce incidence by 84% [11]. While Dye et al. found that health service programmatic indicators did not explain the reduction in tuberculosis incidence [7], Reeves et al. found that reductions in public spending (through economic recession) reduced spending on tuberculosis control and argued that this may lead to increased tuberculosis prevalence [12]. These studies examined associations between different components of economic development and tuberculosis prevalence, but empirical data were limited. In contrast, Siroka et al. used tuberculosis prevalence survey data from 8 countries to examine the association between household-level poverty and tuberculosis prevalence [15]. The study was cross-sectional and did not find a consistent association between household SEP and tuberculosis prevalence. From these studies we therefore understand that it is possible that the relationship between economic development and change in tuberculosis prevalence is not simply dependent on the household or on investment in health services but rather on a combination of risk factors across neighbourhood interactions.

However, economic growth and reduction in poverty may not be the only explanation for the change, as there were also improvements in tuberculosis diagnostics and health service access through an expansion of health insurance in Việt Nam [19,20]. Possible explanations for the results of our study therefore include that the rapid economic development in Việt Nam led to tuberculosis patients being wealthier in the second survey. However, this is mediated by lower participation in the second survey by wealthy households because of the expansion of the Việtnamese National Health Insurance, making the free health check-ups offered for participation in the tuberculosis prevalence survey less attractive [7,12]. Despite lower participation from relatively wealthier households, we found that tuberculosis burden was more concentrated among wealthy households in the 2017 survey than in the 2007 survey, suggesting selective participation.

We found that relative household SEP was weakly associated with tuberculosis prevalence after controlling for known individual-level risk factors such as age and gender [39]. Our finding of a tendency towards tuberculosis being less concentrated among poor households when measured over time corresponds to the findings of Ataguba et al. in South Africa [33]. However, in South Africa, economic development has been accompanied by persistently high levels of income inequality, and the effect was likely mediated by the expansion of the national ART programme, which disproportionally will have benefitted the poor. Our findings suggest that neighbourhood-level (provincial) poverty explains much of the variation in tuberculosis prevalence over time. Neighbourhood-level poverty may be a signal of fewer economic opportunities and therefore a greater vulnerability to tuberculosis [1].

A limitation of our study is the measure of household SEP used, which may have led to the misclassification of household SEP given the limited set of consumption data. Our measure of

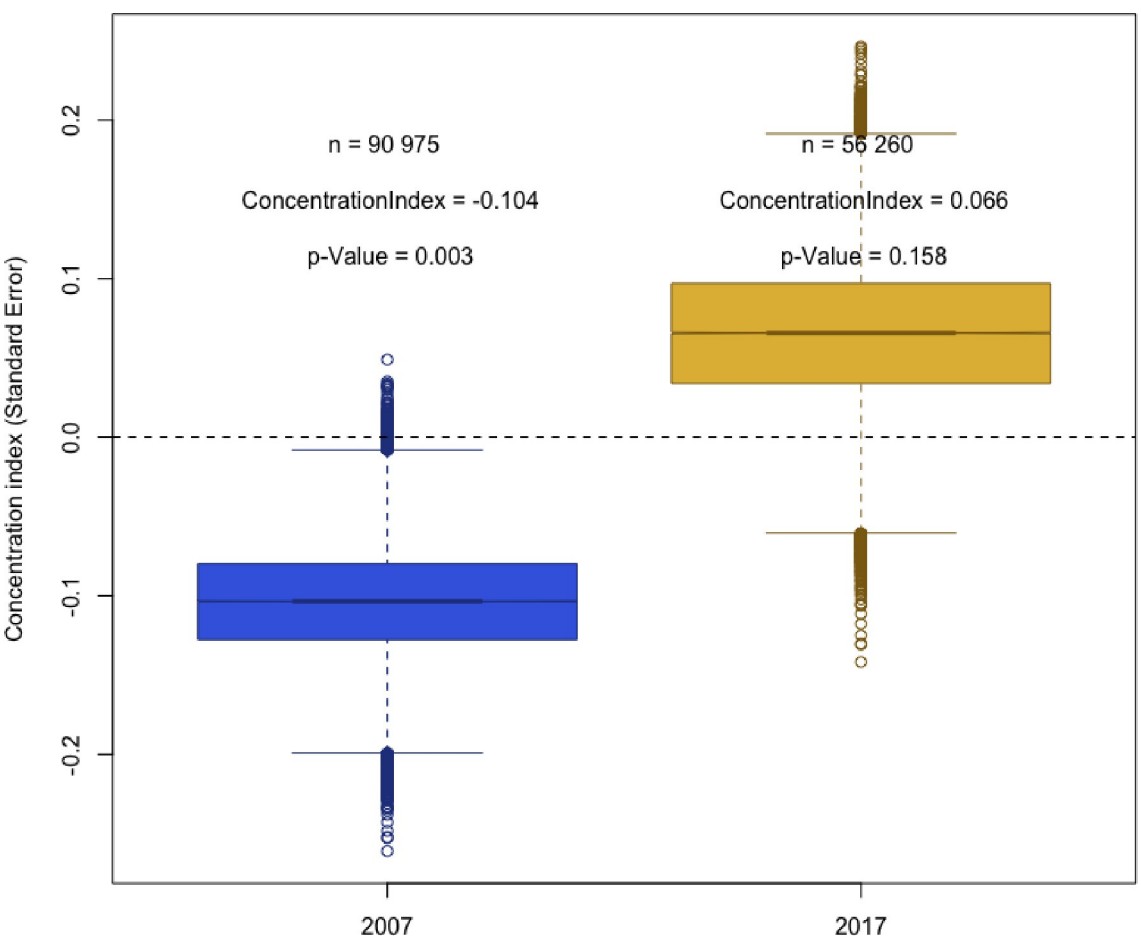

**Fig 5. Illness concentration index for 2007 and 2017 Việt Nam tuberculosis prevalence surveys.** Sampling weights were applied. A negative concentration index means that the health outcome (tuberculosis illness) is concentrated in those who are poor, while a positive index value means that the disease is concentrated in those who are wealthier. The concentration index is an expression of the area between the concentration curve (Fig 4) and the line of perfect equality. In the figure, the bars represent the mean, with the whiskers representing the distribution of data around the mean. The outlying data points are shown as circles above and below the whiskers.

household-level SEP was primarily based on consumption data collected during the prevalence survey, and some of the important factors predicting poverty in Việt Nam such as education were not included in this measure. Furthermore, consumption data are sensitive to change over time; for example, an item that was a signal of prosperity in 2007 may no longer be a good indicator of wealth in 2017. This limitation was mitigated to some extent by using different household- and neighbourhood-level measures of poverty including rural residence, the region where a district is situated in Việt Nam, and the percentage of people in the district who are considered poor (the district poverty rate). We furthermore did not only rely on results based on the consumption-based asset index, but also estimated the absolute wealth of households, and the primary results of the study held across the different measurements used. A further limitation of the study is that while tuberculosis prevalence surveys are valuable for understanding socioeconomic differences in tuberculosis prevalence in countries, given that tuberculosis is a relatively rare disease in the population studied, there is limited power to explore the socioeconomic drivers of tuberculosis prevalence [15]. However, combining repeated cross-sectional surveys with provincial deprivation estimates during a period of

**Table 2. Associations between individual- and household-level variables and tuberculosis prevalence at each timepoint (2007 and 2017).**

| Variable | 2007 survey ($n$ = 94,156) | | 2017 survey ($n$ = 61,763) | |
|---|---|---|---|---|
| | PR (95% CI) | $p$-Value | PR (95% CI) | $p$-Value |
| Age (years) | | | | |
| 15–24 | Ref | | Ref | |
| 25–34 | 1.04 (0.26; 1.81) | 0.009 | 1.19 (−0.06; 2.43) | 0.063 |
| 35–44 | 1.83 (1.12; 2.54) | <0.001 | 1.90 (0.71; 3.08) | 0.002 |
| 45–54 | 2.05 (1.35; 2.76) | <0.001 | 2.10 (0.93; 3.27) | <0.001 |
| 55–64 | 2.36 (1.63; 3.09) | <0.001 | 2.57 (−0.51; 0.62) | <0.001 |
| ≥65 | 2.79 (2.09; 3.49) | <0.001 | 2.81 (1.64; 3.97) | <0.001 |
| Gender | | | | |
| Female | Ref | | Ref | |
| Male | 1.61 (1.29; 1.92) | <0.001 | 1.59 (1.25; 1.92) | <0.001 |
| Region | | | | |
| North | Ref | | Ref | |
| Centre | −0.34 (−0.77; 0.98) | 0.129 | 0.34 (−0.42; 0.72) | 0.081 |
| South | 0.19 (−0.08; 0.47) | 0.170 | 0.64 (0.30; 0.98) | <0.001 |
| Type of residence | | | | |
| Urban | Ref | | Ref | |
| Rural | 0.08 (−0.23; 0.39) | 0.600 | −0.46 (−0.79; −0.14) | 0.005 |
| Remote | −0.16 (−0.53; 0.20) | 0.387 | −0.09 (−0.46; 0.29) | 0.644 |
| Household socioeconomic position | | | | |
| Lowest | Ref | | Ref | |
| Lower middle | −0.22 (−0.54; 0.10) | 0.183 | 0.08 (−0.30; 0.47) | 0.671 |
| Upper middle | 0.19 (−0.17; 0.55) | 0.309 | 0.39 (0.01; 0.76) | 0.042 |
| Highest | −0.41 (−0.81; −0.00) | 0.048 | 0.76 (0.36; 1.16) | <0.001 |

PRs and CIs are estimated using log-binomial mixed effects statistical models. Coefficients are weighted for sampling stratification (differential cluster size, participation by age and gender, stratification by areas and post-stratification weight). CI, confidence interval; PR, prevalence ratio; Ref, reference value; TB, tuberculosis.

remarkable economic growth provides valuable insights into the dynamics of the relationship between tuberculosis and economic development in Việt Nam. Lastly, it is possible that there may have been selection bias due to non-participation in the sampling by individuals during the second survey, in that, with improved economic welfare, there is less incentive for individuals to attend the free screening service provided by the survey, with therefore a bias towards poorer households enrolling. Our findings may therefore be underestimating the true population-level shift in the tuberculosis burden towards wealthier households.

Our results show the potential for tuberculosis prevalence reductions with general and equitable improvements in socioeconomic circumstances in a population. While promoting economic growth, and ensuring that resource distribution is equitable, falls outside the remit of specific national tuberculosis programmes, our study strengthens the case for a multi-sectoral response to tuberculosis [40], which we hope gives further encouragement to policies that aim to achieve this.

## Conclusions

To our knowledge, this is the first study to use repeat direct measurements of tuberculosis burden to empirically examine the relationship between equitable economic development and a reduction in tuberculosis prevalence. We found that with equitable economic growth and a reduction in

**Table 3. Multilevel analyses examining associations between individual-, household-, and neighbourhood-level explanatory variables and change in tuberculosis prevalence.**

| Variable | Model A | | Model B | | Model C | |
|---|---|---|---|---|---|---|
| | PR (95% CI) | *p*-Value | PR (95% CI) | *p*-Value | PR (95% CI) | *p*-Value |
| Tuberculosis prevalence | | | | | | |
| Time (comparator: 2007) | −0.35 (−0.58; −0.12) | 0.003 | −0.35 (−0.69; −0.01) | 0.041 | −0.37 (−0.70; −0.04) | 0.030 |
| **Individual** | | | | | | |
| Age (years) | | | | | | |
| 15–24 | Ref | | Ref | | Ref | |
| 25–34 | 1.34 (0.60; 2.08) | <0.001 | 1.75 (0.97; 2.54) | <0.001 | 1.75 (0.97; 2.53) | <0.001 |
| 35–44 | 1.94 (1.25; 2.64) | <0.001 | 2.26 (1.46; 3.07) | <0.001 | 2.26 (1.46; 3.07) | <0.001 |
| 45–54 | 2.12 (1.42; 2.81) | <0.001 | 2.48 (1.59; 3.38) | <0.001 | 2.48 (1.58; 3.37) | <0.001 |
| 55–64 | 2.41 (1.70; 3.12) | <0.001 | 2.77 (1.93; 3.61) | <0.001 | 2.77 (1.93; 3.61) | <0.001 |
| ≥65 | 2.73 (2.04; 3.42) | <0.001 | 3.01 (2.24; 3.77) | <0.001 | 3.01 (2.24; 3.77) | <0.001 |
| Gender male | 1.42 (1.17; 1.68) | <0.001 | 1.33 (1.06; 1.60) | <0.001 | 1.33 (1.06; 1.60) | <0.001 |
| Region | | | | | | |
| North | | | Ref | | Ref | |
| Centre | | | −0.02 (−0.65; 0.60) | 0.944 | −0.02 (−0.65; 0.61) | 0.950 |
| South | | | 0.24 (−0.20; 0.67) | 0.289 | 0.23 (−0.21; 0.67) | 0.304 |
| Type of residence | | | | | | |
| Urban | Ref | | Ref | | Ref | |
| Rural | −0.13 (−0.38; 0.12) | 0.313 | −0.16 (−0.56; 0.24) | 0.156 | −0.17 (−0.56; 0.23) | 0.410 |
| Remote | −0.28 (−0.63; −0.06) | 0.107 | −0.28 (−0.65; 0.11) | 0.429 | −0.29 (−0.67; 0.09) | 0.138 |
| **Household** | | | | | | |
| Household SEP | | | | | | |
| Lowest | | | | | Ref | |
| Lower middle | | | | | 0.02 (−0.47; 0.51) | 0.931 |
| Upper middle | | | | | 0.25 (−0.05; 0.56) | 0.104 |
| Highest | | | | | 0.23 (−0.19; 0.66) | 0.287 |
| Household AWE | | | 0.004 (−0.00; 0.01) | 0.140 | | |
| **Province** | | | | | | |
| Provincial poverty rate (2009) | −0.01 (−0.02; 0.01) | | 0.11 (0.02; 0.70) | | 0.10 (0.02; 0.68) | |

The dataset includes a total of 155,919 participants. PRs were estimated using mixed effects multilevel models with random intercepts. Model A shows individual-level regressors only, Model B shows individual-level and household-level variables while using the AWE to understand the impact of household wealth, while Model C uses a relative measure of household SEP. Provincial poverty rate is the percentage of the population living below US$2 per day. AWE, absolute wealth estimate; CI, confidence interval; PR, prevalence ratio; Ref, reference value; SEP, socioeconomic position.

tuberculosis burden, tuberculosis became less concentrated among poor households in Việt Nam. The study highlights the important contribution of shared resources to not only reduce poverty but also shifting tuberculosis away from differentially impacting the poorest households.

## Supporting information

**S1 Text. Additional information related to the analysis.**
(DOCX)

**S1 STROBE Checklist. STROBE Checklist.**
(DOC)

## Author Contributions

**Conceptualization:** Nicola Foster, Hai V. Nguyen, Nhung V. Nguyen, Hoa B. Nguyen, Edine W. Tiemersma, Frank G. J. Cobelens, Matthew Quaife, Rein M. G. J. Houben.

**Data curation:** Hai V. Nguyen, Matthew Quaife.

**Formal analysis:** Nicola Foster, Rein M. G. J. Houben.

**Funding acquisition:** Rein M. G. J. Houben.

**Investigation:** Hai V. Nguyen, Nhung V. Nguyen, Hoa B. Nguyen, Edine W. Tiemersma, Frank G. J. Cobelens, Matthew Quaife, Rein M. G. J. Houben.

**Methodology:** Nicola Foster, Frank G. J. Cobelens.

**Supervision:** Nhung V. Nguyen, Hoa B. Nguyen, Edine W. Tiemersma, Frank G. J. Cobelens, Matthew Quaife, Rein M. G. J. Houben.

**Writing – original draft:** Nicola Foster.

**Writing – review & editing:** Nicola Foster, Hai V. Nguyen, Nhung V. Nguyen, Hoa B. Nguyen, Edine W. Tiemersma, Frank G. J. Cobelens, Matthew Quaife, Rein M. G. J. Houben.

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
