## [Editor Report · Decision Letter 0]

28 Jul 2021

Dear Dr Foster, 

Thank you for submitting your manuscript entitled "Reducing the burden of tuberculosis among the poor: social determinants of the changing tuberculosis prevalence in Viet Nam" for consideration by PLOS Medicine.

Your manuscript has now been evaluated by the PLOS Medicine editorial staff and I am writing to let you know that we would like to send your submission out for external peer review.

Please re-submit your manuscript within two working days, i.e. by Jul 30 2021 11:59PM.

Kind regards,

Beryne Odeny

Associate Editor

PLOS Medicine

---

## [Decision Letter · Decision Letter 1]

7 Oct 2021

Dear Dr. Foster,

Thank you very much for submitting your manuscript "Reducing the burden of tuberculosis among the poor: social determinants of the changing tuberculosis prevalence in Viet Nam" (PMEDICINE-D-21-03248R1) for consideration at PLOS Medicine. 

[LINK]

In light of these reviews, I am afraid that we will not be able to accept the manuscript for publication in the journal in its current form, but we would like to consider a revised version that addresses the reviewers' and editors' comments. Obviously we cannot make any decision about publication until we have seen the revised manuscript and your response, and we plan to seek re-review by one or more of the reviewers. 

We expect to receive your revised manuscript by Oct 28 2021 11:59PM. Please email us (plosmedicine@plos.org) if you have any questions or concerns.

We look forward to receiving your revised manuscript. 

Sincerely,

Beryne Odeny, 

PLOS Medicine

plosmedicine.org

1) Please revise your title according to PLOS Medicine's style. Your title must be nondeclarative and not a question. It should begin with main concept if possible. Please place the study design in the subtitle (i.e., after a colon). For example, “Changing tuberculosis prevalence by socioeconomic status in Viet Nam: Analysis of population-level cross-sectional studies.” 

2) The Data Availability Statement (DAS) requires revision. Please include an appropriate weblink to the LSHTM Data Compass website.

3) Abstract:

a) Please structure your abstract using the PLOS Medicine headings (Background, Methods and Findings, Conclusions).

b) Please combine the Methods and Findings sections into one section, “Methods and findings”. 

c) Please ensure that all numbers presented in the abstract are present and identical to numbers presented in the main manuscript text.

d) Please include the important dependent variables that are adjusted for in the analyses.

e) In the last sentence of the Abstract Methods and Findings section, please describe the main limitation(s) of the study's methodology.

4) Please place the Author Summary after the Abstract.

5) Did your study have a prospective protocol or analysis plan? Please state this (either way) early in the Methods section:

 a) If a prospective analysis plan (from your funding proposal, IRB or other ethics committee submission, study protocol, or other planning document written before analyzing the data) was used in designing the study, please include the relevant prospectively written document with your revised manuscript as a Supporting Information file to be published alongside your study and cite it in the Methods section. A legend for this file should be included at the end of your manuscript. 

c) In either case, changes in the analysis - including those made in response to peer review comments-- should be identified as such in the Methods section of the paper, with rationale.

6) Please ensure that the study is reported according to the STROBE guideline for observational studies, and include the completed STROBE checklist as Supporting Information. Please add the following statement, or similar, to the Methods: "This study is reported as per the Strengthening the Reporting of Observational Studies in Epidemiology (STROBE) guideline (S1 Checklist)." The STROBE guideline can be found here: http://www.equator-network.org/reporting-guidelines/strobe/

7) Your study is observational and therefore causality cannot be inferred. Please remove language that implies causality throughout the manuscript, such as “determinants”, “contributed to”, “explanation for”, and so forth. Refer to associations instead.

8) Please avoid assertions of primacy ("We report for the first time....") by stating, “to our knowledge” or similar

9) Please clarify whether population case detection rates, treatment success rates, or earlier diagnosis changed during the study period in the country and may have affected prevalence.

10) Did you account for population changes in your multilevel models? 

11) Figures and tables

a) Please include the total N at the top of table 2 & 3

b) Please define abbreviations such in the table footnotes. For example, SEP, AWE, CI

c) Please provide a meaning for the bars and whiskers in figures 2 and 5.

12) The terms gender and sex are not interchangeable (as discussed in http://www.who.int/gender/whatisgender/en/ ); please use the appropriate term.

13) In the Discussion, please include the limitations of the study, implications and next steps for research, clinical practice, and/or public policy.

14) References:

a) Please ensure in-text reference call outs are enclosed in square brackets. For example, “... countries [1,2]."

b) Please include access dates for all weblinks (e.g., Ref #14, 26), and ensure that all weblinks are current and accessible.

c) Please update ref #23 or delete if not published.

15) To help us extend the reach of your research, please provide any Twitter handle(s) that would be appropriate to tag, including your own, your coauthors’, your institution, funder, or lab.

Comments from the reviewers:

Reviewer #1: Dear authors,

Firstly, I congratulate the authors for the dedication and focus to fortify the science in tuberculosis thematic area. The manuscript is important to the public health and provides high importance to better understanding the impact of socio-economic factors on decrease the burden of tuberculosis, and in this way, for health decision making. 

I strongly suggest to the authors to follow the PLOS Medicine's submission guidelines. Please use continuous line numbers, do not restart the numbering on each page. In the text, cite the reference number in square brackets according the PLOS Medicine's submission guidelines. 

Please, organize the level 1 and 2 heading in the manuscript body according to https://journals.plos.org/plosmedicine/s/file?id=9cba/PLOS%20Manuscript%20Body%20Formatting%20Guidelines.pdf

I suggest reading the manuscript carefully, since there are many little mistakes that could be solved easily by the authors. 

The manuscript requires professional editing in the English language to make it easy to read and understand.

I strongly suggest to the authors the Strengthening the Reporting of Observational Studies in Epidemiology (STROBE) Statement: Guidelines for Reporting Observational Studies. 

It is available from von Elm E, Altman DG, Egger M, Pocock SJ, Gøtzsche PC, Vandenbroucke JP, et al. (2007) The Strengthening the Reporting of Observational Studies in Epidemiology (STROBE) Statement: Guidelines for Reporting Observational Studies. PLoS Med 4(10): e296. https://doi.org/10.1371/journal.pmed.0040296

I also suggest to the authors to verify the United to End TB: Every Word Counts launched by Stop TB Partnership. Available from: http://www.stoptb.org/assets/documents/resources/publications/acsm/LanguageGuide_FtorWeb20131110.pdf

The language guide supports the call for change in the upcoming Global Plan to End TB 2016-2020, which includes changing the mindset, language, and dialogue on TB as one of the key paradigm shifts required to reach the End TB goals.

Title page

There are unnecessary information in the title page. Please, follow the PLOS Medicine's submission guidelines (https://journals.plos.org/plosmedicine/s/file?id=3fac/PLOS%20Affiliations%20Formatting%20Guidelines.pdf). 

Line 21 - 23: Delete these information 

Line 32 - 36: Delete these information

Line 1 - 3: Delete these information

Line 5: Key words are not necessary according to the PLOS Medicine's submission guideline. 

Title

Line 2: Please, replace the word "Viet Nam" by "Vietnam", according to the English spelling in the whole manuscript body. 

I suggest to the authors the title "Social determinants and changing prevalence in decrease of the burden of tuberculosis in Vietnam". 

Author summary

Page 3: Please delete the author summary heading. This section is not necessary in the manuscript and it does not follow the PLOS Medicine's submission guidelines. Merge this information in the background section, methods section, discussion section and conclusion section. 

Abstract

The authors must follow the submission guidelines to provide a better abstract of the study. I suggest to the authors to minimize the use of abbreviations. 

Line 10: Please, replace the word ˜Methods" by "Methods and findings". 

Line 27: Please, replace the word "interpretation" by "conclusion" 

Line 31: Please, delete the "funding" heading. 

Background

The Background section should explain the background to the study, its aims, a summary of the existing literature and why this study was necessary or its contribution to the field. There are many blanks about the existing literature about the theme in the manuscript. 

I strongly suggest an explanation how the Social Determinants of Health impact the prevalence and how they increase the burden of tuberculosis. A deep literature review about the theme is necessary to understand the contribution to the public health field. 

Line: 2: Replace the word "Introduction" by "Background".

Line 4: Replace the abbreviation "TB" after the word "tuberculosis".

Line 5: The authors stated: "Ecological studies…", but the authors just cited one reference. Please, insert references in this statement. 

Line 7 - 10: Please, rewrite this statement and insert a reference. 

Line 24: Replace the word "Viet Nam" by "Vietnam". 

Line 33 - 34: Please insert a reference in this statement. 

Line 1 - 2: Please insert a reference in this statement.

Line 2 - 5: Rewrite the paragraph adding the aim of the study following the scientific writing criteria (verb in the infinite tense). 

Methods 

The authors must follow the STROBE statement to better report the sections in the manuscript body, specially in the methods section.

Add new headings according to the STROBE statement (study design, setting…)

The paragraphs do not have a logical sequential that allow to understand the aim of the study. 

Line 23: TB is not associated only with the crowding places. Rewrite this statement and insert a reference. 

Please, organize the level 2 heading in the methods section according to https://journals.plos.org/plosmedicine/s/file?id=9cba/PLOS%20Manuscript%20Body%20Formatting%20Guidelines.pdf

Line 20: Move the meaning of the abbreviation for the first time it appears. 

In the Statistical Analysis section, why did the authors only use six variables to asset indices? 

Results

The results section is well reported. There are major revisions in this section. 

The results reported in this section are not clear. I suggest rearranging this section to make it clearer and more readable according to the STROBE Statement. 

I suggest to the authors to use the comma to separate the number every third digit from the right.

Discussion

The discussion section is well reported. There are major revisions in this section. 

I suggest to the authors a discussion based in the data reported in the results section and, how the findings can subsidize the policy makers in different scenarios to obtain the best outcomes in decrease TB burden. 

I suggest to the authors to insert the limitations of the study in the last paragraphs in the discussion section. 

Please, discuss the data from survey one and two. Is there a point of integration or similarity? 

Line 11 - 13: Please include a reference in this statement. 

Line 17: Insert a point after the letter "l" in the "at al."

Line 18: Reference correctly the citation "Carter at al.". 

Line 35 - 36: Please include a reference in this statement.

Line 10: Insert a point after the letter "l" in the "at al."

Conclusion

Please include in the conclusion section how this study will support, for example, policy makers to better understanding the impact of socio-economic factors on increase the burden of tuberculosis orientating the efforts to accomplish the aims of The End TB strategy. 

References

In the text, cite the reference number in square brackets according the PLOS Medicine's submission guidelines. 

I suggest to the authors to insert some other references to be included in the discussion section. These studies were carried out in different scenarios and could improve this study. 

1. https://doi.org/10.1371/journal.pone.0249822

2. https://doi.org/10.5588/ijtld.18.0149

3. http://dx.doi.org/10.2471/BLT.06.038331

4. https://doi.org/10.1016/j.jiph.2020.03.010

5. https://doi.org/10.1016/S1473-3099(09)70041-6

6. https://doi.org/10.1111/tmi.13409

Reviewer #2: Well done to the team. My comments are in the attached manuscript. 

Reviewer #3: Congratulation for the excellent work and the well written manucript

Reviewer #4: I confine my remarks to statistical aspects of this paper.

First, there is the general issue of causal language. This is an observational study and causality cannot be inferred. Even in the title the word "determinants" is used. Not only should causal words be changed, but the "causal pathways" section should make it clear that these are *possible* causal pathways.

p. 9 line 4 This should really be factor analysis, not PCA. The results are often similar, but the goals are different. The goals here are clearly those of factor analysis - to uncover a latent factor from several measures that are related to it. 

 line 8-9 Why were these categorized? Categorizing a continuous variable is nearly always a mistake. The index can be used as is. Nonlinearity can be explored with splines. If the authors then want to compare various percentiles, they can do so. 

 line 15 Please explain what an "illness concentration curve" is.

p. 11 line 17-19 The two measures are also not inconsistent. But these results are inconsistent with the GINI results cited earlier. 

Table 2 Do not categorize age or SES. See my blog post https://medium.com/@peterflom/what-happens-when-we-categorize-an-independent-variable-in-regression-77d4c5862b6c

The AIC is of no use here. It is only useful for comparing different models on the same data set.

Peter Flom

[LINK]

---

## [Decision Letter · Decision Letter 2]

24 Jan 2022

Dear Dr. Foster,

Thank you very much for re-submitting your manuscript "Social determinants of the changing tuberculosis prevalence in Việt Nam: analysis of population-level cross-sectional studies" (PMEDICINE-D-21-03248R2) for review by PLOS Medicine.

I have discussed the paper with my colleagues and the academic editor and it was also seen again by two reviewers. I am pleased to say that provided the remaining editorial and production issues are dealt with we are planning to accept the paper for publication in the journal.

[LINK]

We look forward to receiving the revised manuscript by Jan 24 2022 11:59PM.   

Sincerely,

Beryne Odeny, 

PLOS Medicine

plosmedicine.org

Requests from Editors:

1) Thank you for providing a link to the description of the dataset; however, this link is inactive and DOI cannot be found.

2) Please situate the Author Summary after the Abstract.

3) In the last sentence of the Abstract’s Methods and Findings section, please describe the main limitation(s) of the study's methodology.

4) In the Abstract conclusion, please include a sentence on implications and next steps for clinical practice and public policy.

5) Thank you for providing your STROBE checklist. Please replace the page numbers with paragraph numbers per section (e.g. "Methods, paragraph 1"), since the page numbers of the final published paper may be different from the page numbers in the current manuscript.

6) References:

a) Please include access dates for all weblinks (e.g., Ref #32, and ensure that all weblinks are current and accessible.

b) Please reformat the citation style into PLOS Medicine's format and ensure journal name abbreviations consistently match those found in the National Center for Biotechnology Information (NCBI) databases

Comments from Reviewers:

Reviewer #1: Dear authors, 

Thank you for revising the manuscript considering the suggestions.

The manuscript is now technically sound. 

I do not have additional comments on the revision in this manuscript version.

Reviewer #4: The authors have addressed my concerns and I now recoommend publication.

Peter Flom

[LINK]

---

## [Editor Report · Decision Letter 3]

3 Feb 2022

Dear Dr Foster, 

On behalf of my colleagues and the Academic Editor, Dr. Amitabh Bipin Suthar, I am pleased to inform you that we have agreed to publish your manuscript "Social determinants of the changing tuberculosis prevalence in Việt Nam: analysis of population-level cross-sectional studies" (PMEDICINE-D-21-03248R3) in PLOS Medicine.

PRESS

Sincerely, 

Beryne Odeny 

PLOS Medicine